

# Numerical simulation of mantle convection using a temperature dependent nonlinear viscoelastic model

By *M. Norouzi*[*,1], *M. Sheibi*[2] and *M. Mahmoudi*[3]

[1, 3] Mechanical Engineering Department, Shahrood University of Technology, Shahrood, Iran
[2] Earth Science Faculty, Shahrood University of Technology, Shahrood, Iran

## ABSTRACT

In the present article, the mantle convection is simulated numerically using a temperature dependent non-linear viscoelastic model for the first time. The numerical domain of problem is considered as a 4000km*2000km rectangular box and the CFD simulation is performed using finite volume method. Unlike the previous works which had been investigated the mantle convection using the linear viscoelastic models or simple nonlinear inelastic viscous equations (such as power law or cross equations), it is solved via the nonlinear Giesekus constitutive equation. Because of large-scale creeping flow in geometry and time, it is shown that the results of Giesekus equation are more reliable for this problem. The main innovative aspects of current study is investigation of temperature dependency of rheological properties of mantle including viscosity, normal stress differences and relaxation time using appropriate equations of state. The variation of gravitational acceleration with depth of Earth and the effect of the work of stress field (viscous dissipation) on mantle convection are also simulated for the first time.

*Keywords:* Mantle convection; Giesekus model; Numerical simulation; Temperature dependence rheological properties.



**Nomenclature**

| Parameter | Symbol | Units |
|---|---|---|
| Brinkman number | $Br$ | |
| Heat Capacity | $C_p$ | J kg$^{-1}$K$^{-1}$ |
| Elastic number | $En$ | |
| Gravity acceleration | $g$ | m s$^{-2}$ |
| Depth of mantle | $H$ | km |
| Thermal conductivity | $k$ | Wm$^{-1}$K$^{-1}$ |
| Nusselt number | $Nu$ | |
| Pressure | $p$ | pa |
| Prantdl number | $Pr$ | |
| Rayleigh number | $Ra$ | |
| Reynolds number | $Re$ | |
| Time | $t$ | Gyr |
| Temperature | $T$ | K |
| Velocity vector | $U$ | mm yr$^{-1}$ |
| Reference velocity | $W_0$ | mm yr$^{-1}$ |
| Weissenberg number | $We$ | |

**Greek Symbols**

| | | |
|---|---|---|
| Mobility factor | $\alpha$ | |
| Compressibility factor | $\beta_C$ | Mpa$^{-1}$ |



| Viscosity ratio | $\beta_G$ | |
| Thermal expansivity | $\beta_T$ | $K^{-1}$ |
| Stress field work | $\Phi$ | |
| Shear rate | $\dot{\gamma}$ | $s^{-1}$ |
| Exponential rate | $\Gamma$ | $K^{-1}$ |
| Dynamic viscosity | $\eta$ | $kg\ m^{-1}\ s^{-1}$ |
| Thermal diffusivity | $\kappa$ | |
| Relaxation time | $\lambda$ | $s$ |
| Kinetic viscosity | $\nu$ | $m^2\ s^{-1}$ |
| Density | $\rho$ | $kg\ m^{-3}$ |
| Stress tensor | $\tau$ | pa |

**Subscripts**

| Property at upper plate | $0$ |
| Newtonian | $n$ |
| Viscoelastic | $v$ |

**1. INTRODUCTION**
Mantle convection is a creeping flow in the mantle of the Earth that causes some
convective currents in it and transfers heat between core and Earth's surface. In fluid
mechanics, the free convection is a classic topic driven by the effect of temperature
gradient on density. This solid-state convection in mantle is an abstruse phenomenon



that carries out various tectonic activities and continental drift (Bénard (1900),
Batchelor (1954), Elder (1968)). This motion occurs on a large scale of space and time.
From fluid mechanics point of view, mantle convection is approximately a known
phenomenon; the only force which causes convective flow is buoyancy force while this
phenomenon is affected by the nature of non-Newtonian rheology (Christensen (1985))
and depth-and temperature-dependent viscosity. Gurnis and Davies (1986) just used a
depth dependent viscosity and assumed that the Rayleigh number is constant. They
deduced this phenomenon depend on Rayleigh number, as when $Ra$ is increased, the
thermal boundary layer will be thinned and the center of circulation shifts more to the
narrow descending limb. Hansen *et al.* (1993) examined the influences of both depth-
dependent viscosity and depth-dependent thermal expansivity on the structure of mantle
convection using two-dimensional finite-element simulations. They concluded depth-
dependent properties encourage the formation of a stronger mean flow in the upper
mantle, which may be important for promoting long-term polar motions. The rheology
of mantle strongly depends on the temperature and hydrostatic pressure (Ranalli (1995),
Karato (1997)). Also, because of huge geometry of Earth's mantle (2000km), the
gravity cannot be considered as a constant, and it is a function of depth.
Kellogg and King (1997) developed a finite element model of convection in a
spherical geometry with a temperature-dependent viscosity. They have focused on three
different viscosity laws: (1) constant viscosity, (2) weakly temperature-dependent
viscosity and (3) strongly temperature-dependent viscosity. Moresi and Solomatov
(1995) have simulated it as two-dimensional square cell with free-slip boundaries. They



reached an asymptotic regime in the limit of large viscosity contrasts and obtained
scaling relations that found to be agreement with theoretical predictions. Ghias and
Jarvis (2008) investigated the effects of temperature- and depth-dependent thermal
expansivity in two-dimensional mantle convection models. They found the depth and
temperature dependence of thermal expansivity each have a significant, but opposite,
effect on the mean surface heat flux and the mean surface velocity of the convective
system. The effect of temperature-dependent viscosity was studied in literature in two-
dimensional rectangular domains (Severin and Herwig (1999), Pla *et al.* (2009),
Hirayama and Takaki (1993), Fröhlich *et al.* (1992)). Tomohiko *et al.* (2004) simulated
a two-dimensional rectangular domain with assuming the mantle as an incompressible
fluid with a power-law viscosity model. They employed a simplified two-layer
conductivity model and studied the effects of depth-dependent thermal conductivity on
convection using two-dimensional Boussinesq convection model with an infinite
Prandtl number. Their results implied that the particular values of thermal conductivity
in horizontal boundaries could exert more significant influence on convection than the
thermal conductivity in the mantle interior. Stein *et al.* (2004) explored the effect of
different aspect ratios and a stress- and pressure-dependent viscosity on mantle
convection using three-dimensional numerical simulation. Ozbench *et al.* (2008)
presented a model of large-scale mantle-lithosphere dynamics with a temperature-
dependent viscosity. Ichikawa *et al.* (2013) simulated a time-dependent convection of
fluid under the extended Boussinesq approximation in a model of two-dimensional
rectangular box with a temperature- and pressure-dependent viscosity and a viscoplastic
property. Stien and Hansen (2008) employed a three-dimensional mantle convection



model with a strong temperature, pressure and stress dependence of viscosity and they
used a viscoplastic rheology. Kameyama and Ogawa (2000) solved thermal convection
of a Newtonian fluid with temperature-dependent viscosity in a two-dimensional
rectangular box. Kameyama *et al.* (2008) considered a thermal convection of a high
viscous and incompressible fluid with a variable Newtonian viscosity in a three-
dimensional spherical geometry. Gerya and Yuen (2007) simulated a two-dimensional
geometry and non-Newtonian rheology using power-law model.

83       In the present paper, the mantle convection is simulated numerically using a

temperature dependent non-linear viscoelastic model for the first time. The geometry of
problem is shown in Fig. 1. Here, the calculation domain is considered as a
4000km×2000km rectangular box. Two hot and cold plates are considered at the bottom
and top of box, respectively. The isolator thermal condition is considered at the left and
right hand sides of domain. The problem is solved via a second order finite volume
method. The effect of temperature on rheological properties consist of the viscosity,
normal stress differences and relaxation time of mantle are modeled using appropriate
equations of state which are the main innovative aspects of current study. The variation
of gravitational acceleration with depth of Earth and the effect of the work of stress field
(viscous dissipation) on mantle convection are simulated for the first time. According to
the literature, the previous studies are restricted to the linear and quasi-linear
viscoelastic constitutive equations and the nonlinearity nature of mantle convection was
modeled as simple nonlinear constitutive equations just for apparent viscosity such as
the power-law and cross models. Here, the Giesekus nonlinear viscoelastic model is



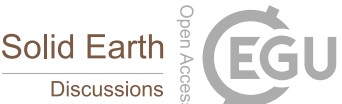

used as the constitutive equation. This high order nonlinear model is used because of
large-scale creeping viscoelastic flow of mantle convection in domain and time. Using
Giesekus constitutive equation, we can calculate a more accurate solution for this
problem because:
1. In addition to the viscosity, the shear dependencies of other viscometric
functions (consist of the first and second normal stress differences) are also
modeled. It is important to remember that the linear and quasi-linear viscoelastic
constitutive equations that used in previous studies could not able to model the
completed set of shear dependent nonlinear viscometric functions which resulted
from anisotropic behavior of flow field.
2. The effect of the third invariant of shear rate tensor on stress field (especially for
normal stress components) is also modeled for the first time. The simple non-
linear viscous models such as power-law and cross equations that used in
previous studies depend only on generalized shear rate which is defined based
on the second invariant of the shear rate.
3. The nonlinear effect of material elasticity on large deformation of mantle is
modeled simultaneity with the effects of viscometric functions and elongational
rheological properties.
4. It is important to remember that the non-linear constitutive equations like as the
Giesekus equation could able to model the material elasticity and relaxation
spectra much better than linear models for large deformations of flow field.


**2. GOVERNING EQUATIONS**
The governing equations of an incompressible viscoelastic fluid flow consist of the
continuity, momentum and energy equations:

$$\nabla.\tilde{U} = 0 \tag{1a}$$

$$\rho\frac{\partial \tilde{U}}{\partial \tilde{t}} + \rho\tilde{U}.\nabla\left(\tilde{U}\right) = -\nabla\tilde{p} + \nabla.\tilde{\tau} + \rho\boldsymbol{g} \tag{1b}$$

$$\rho c\left(\frac{\partial \tilde{T}}{\partial \tilde{t}} + \tilde{U}.\nabla\tilde{T}\right) = \nabla.\left(k\nabla\tilde{T}\right) + \tilde{\tau}:\nabla\tilde{U} + \tilde{u}''' \tag{1c}$$

where $\tilde{U}$ is the velocity vector, $\rho$ is density, $c$ is heat capacity, $\tilde{p}$ is static pressure, $\tilde{T}$
is temperature, $k$ is thermal conductivity, $\tilde{t}$ is time, $\tilde{u}'''$ is power of heat source and $\tilde{\tau}$ is
the total stress tensor. The stress tensor is consisted as the summation of Newtonian $\tilde{\tau}_n$
and viscoelastic contributions $\tilde{\tau}_v$ as follows:

$$\tilde{\tau} = \tilde{\tau}_n + \tilde{\tau}_v \tag{2}$$

In Newtonian law ($\tilde{\tau}_n = \tilde{\eta}_n\tilde{\dot{\gamma}}$), $\tilde{\eta}_n$ and $\tilde{\dot{\gamma}}$ which respectively are the constant solvent
viscosity and the shear rate tensor, gives the solvent part $\tilde{\tau}_n$. The viscoelastic stress will
be obtained from a constitutive equation. The usefulness of a constitutive equation for
describing processing flows of viscoelastic solutions and melts rest on its ability to
accurately predict rheological data, as well as on its numerical tractability in several
flow geometries. Such equation should successfully account for shear dependent



viscosity, normal stress effects in steady shear flows, elastic effects in shear-free flows
and non-viscometric flow phenomena. The parameter $\beta_G$ represents the relation of
viscoelastic behavior (as the additives) with pure Newtonian behavior (as the solvent):

$$\beta_G = \frac{\tilde{\eta}_v}{\tilde{\eta}_n + \tilde{\eta}_v} \qquad (3)$$

Since the present study examines mantle convection, this parameter must be near unity.
In other words, the viscoelastic portion dominates to pure Newtonian portion in
behavior of fluid flow. Therefore, the main portion of viscosity of mantle could be
attributed to the $\tilde{\eta}_v$.
The Giesekus model is a popular choice, because of its relative success in several
flows, and its reduction to several well-known simpler models, which make it useful in
a variety of flow situations. The key characteristic of this model is that it includes non-
linear term in stress. Here, the Giesekus model is used as the non-linear constitutive
equation:

$$\tilde{\boldsymbol{\tau}}_v + \lambda \tilde{\boldsymbol{\tau}}_{v(1)} + \alpha \frac{\lambda}{\tilde{\eta}_v}\left(\tilde{\boldsymbol{\tau}}_v.\tilde{\boldsymbol{\tau}}_v\right) = \tilde{\eta}_v \tilde{\dot{\gamma}} \qquad (4)$$

where $\tilde{\eta}_v$ is the viscosity contribution of viscoelastic material at zero shear rate and $\tilde{\boldsymbol{\tau}}_{v(1)}$
is the upper convected derivative of viscoelastic stress tensor defined by:

$$\tilde{\boldsymbol{\tau}}_{v(1)} = \frac{D}{D\tilde{t}}\tilde{\boldsymbol{\tau}}_v - \nabla \tilde{U}^T.\tilde{\boldsymbol{\tau}}_v - \tilde{\boldsymbol{\tau}}_v.\nabla \tilde{U} \qquad (5)$$





in which $\dfrac{D(\ )}{D\tilde{t}}$ is material derivative operator given by $\dfrac{D(\ )}{D\tilde{t}} = \dfrac{\partial(\ )}{\partial\tilde{t}} + \tilde{U}.\nabla(\ )$. The
Giesekus constitutive equation is derived by kinetic theory, arising naturally for
polymer solutions. This model contains four parameters: a relaxation time $\lambda$; the solvent
and polymeric contributions at the zero-shear rate viscosity, $\tilde{\eta}_n$ and $\tilde{\eta}_v$; and the
dimensionless "mobility factor" $\alpha$ (Bird *et al.* (1987)). The origin of the term involving
$\alpha$ can be associated with anisotropic Brownian motion and/or anisotropic
hydrodynamic drag on the constitutive of heavy particles.
In this paper, the viscosity is assumed to be depended on depth and temperature as
follow:

$$\tilde{\eta} = \tilde{\eta}_0 \exp\left[1.535|y| - \Gamma\left(\tilde{T} - \tilde{T}_0\right)\right] \tag{6}$$

where $\tilde{\eta}_0$ is the total viscosity at reference temperature $(T_0)$, $y$ is the depth (per
1000Km), and $\Gamma$ is the exponential rate. The relaxation time $(\lambda)$ is also assumed to be
an exponential function of temperature:

$$\lambda = \lambda_0 \exp\left[-\Gamma\left(\tilde{T} - \tilde{T}_0\right)\right] \tag{7}$$

Because of large scale of geometry and the nature of mantle convection, the dependency
of density on temperature and pressure are considered as follows:

$$\rho = \rho_0\left[1 - \kappa\left(\tilde{T} - \tilde{T}_0\right)\right]\left[1 + \beta_C\left(\tilde{p} - \tilde{p}_0\right)\right] \tag{8}$$



where $\tilde{T}_0 = 300K$ and $\tilde{p}_0 = 0.1MPa$ are reference temperature and pressure,
respectively, $\rho_0$ is density at reference temperature and pressure, $\kappa$ is thermal
expansivity and $\beta_C$ is compressibility coefficient.

**3. NON-DIMENSIONALIZATION**
According to Fig. 1, the Cartesian coordinate system is used in this study. The
dimensionless parameters of flow field are as follows:

$$x = \frac{\tilde{x}}{H} \qquad\qquad y = \frac{\tilde{y}}{H} \qquad\qquad \boldsymbol{U} = \tilde{\boldsymbol{U}} / W_0$$

$$\boldsymbol{\tau} = \frac{\tilde{\tau}H}{\tilde{\eta}_0 W_0} \qquad\qquad p = \frac{\tilde{p}H}{\tilde{\eta}_0 W_0} \qquad\qquad \eta = \frac{\tilde{\eta}}{\tilde{\eta}_0} \qquad\qquad (9)$$

$$Re = \frac{\rho W_0 H}{\tilde{\eta}_0} \qquad\qquad We = \frac{\lambda W_0}{2H} \qquad\qquad En = \frac{We}{Re}$$

where $\tilde{x}$ and $\tilde{y}$ are indicating the coordinate directions; $H$ is the depth of geometry, $W_0$
is the reference velocity, $\tilde{\eta}_0$ is the dynamic viscosity at zero shear rate ($\tilde{\eta}_0 = \tilde{\eta}_v + \tilde{\eta}_n$), $\tilde{\eta}$
is the fluid viscosity, $\rho$ is density and *Re*, *We* and *En* are the Reynolds, Weissenberg
and Elastic numbers, respectively. The ~ notation signifies that parameter has
dimension. The governing dimensionless parameters of heat transfer are as follows:

$$T = \frac{\tilde{T} - \tilde{T}_{min}}{\tilde{T}_{max} - \tilde{T}_{min}} \qquad\qquad Br = \frac{\eta_0 W_0^2}{k\left(\tilde{T}_{max} - \tilde{T}_{min}\right)} \qquad\qquad Pr = \frac{\eta_0}{\rho\kappa} \qquad\qquad (10)$$





$$Ra = \frac{g\beta_T \Delta \tilde{T} H^3}{\nu^2} Pr \qquad Nu = \frac{hH}{k}$$

In the above relations, $T$ is the dimensionless temperature; $\tilde{T}_{min}$ and $\tilde{T}_{max}$ are the
minimum and maximum temperature of fluid, respectively; $k$ is the conduction
coefficient, $\kappa$ is thermal diffusivity, $h$ is the convection heat transfer coefficient and
$Br$, $Pr$, $Ra$ and $Nu$ are the Brinkman, Prandtl, Rayleigh and Nusselt numbers,
respectively. Thus, the dimensionless form of continuity and momentum equations are
as follows:

$$\nabla \cdot \boldsymbol{U} = 0 \tag{11a}$$

$$\boldsymbol{U} \cdot \nabla \boldsymbol{U} = \frac{g\beta \Delta \tilde{T} H}{W_0^2} T + \frac{1}{Re} \nabla^2 \boldsymbol{U} \tag{11b}$$

$$\boldsymbol{U} \cdot \nabla T = \frac{1}{RePr} \left\{ \nabla \cdot (\nabla T) + Br\Phi \right\} \tag{11c}$$

where $\beta$ is the thermal expansion coefficient. In order to get closer to reality, in the
energy equation, we assume a viscosity dissipation term ($\boldsymbol{\tau} : \nabla \times \boldsymbol{U}$). This term is the
effect of stress field work on fluid flow and for Newtonian fluids; it has always a
positive sign according to the second law of thermodynamic. Actually, this positive
term refer to the irreversibility of flow field work and thus in Newtonian fluid it is
known as viscosity dissipation. The interesting point of this term for viscoelastic fluids
is the local possibility of being negative. In effect, having locally negative value of this
term indicates that part of energy is saved in elastic constituent of fluid (Bird *et al.*





(2002)). In Eq. (11c), $\Phi$ is the dimensionless form of work of stress field and obtain
from following equation:

$$\Phi = \tau_{xx} \frac{\partial U_1}{\partial x} + \tau_{xy} \left( \frac{\partial U_1}{\partial y} + \frac{\partial U_2}{\partial x} \right) + \tau_{yy} \frac{\partial U_2}{\partial y} \qquad (12)$$

This variation in viscosity introduces a relativity factor in the problem. Here, the non-
dimensionalization is performed regarding to the value of the viscosity in the upper
plate. Therefore, a new Rayleigh number should be defined, due to the variation of
viscosity: $Ra_{new} = Ra \exp\left(-\Gamma\left(T - T_0\right)\right)$.

193         In our numerical calculations, the values of the parameters are related to the values in

the mantle (Pla *et al.*, 2010), Table 1 shows the values of parameters used in
calculations. Due to the nature of mantle convection the *Pr* number and viscosity are
assumed to be in order of $10^{26}$ and $10^{20}$, respectively. Also, a Rayleigh number equal to
227 is used for this simulation.

198         Remember that the gravitational acceleration of the Earth is decreased by increasing

the depth. Because of the large scale of geometry, the variation of gravitational
acceleration with depth is considered in present study. For this purpose, we used the
data of Bullen (1939) and fitted the following six order interpolation on them with 95%
confidence:

$$g\left(y\right) = -0.118 y^6 + 0.602 y^5 - 1.006 y^4 + 0.6884 y^3 - 0.3708 y^2 + 0.167 y - 9.846 \qquad (13)$$



where $y$ ($1000 Km$) is the depth from bottom plate. We used the above equation in CFD
simulation of mantle convection which is the other innovative aspect of present study.

**3. NUMERICAL METHOD, BOUNDARY AND INITIAL CONDITIONS**
There are totally eight solution variable parameters in the discretized domains,
comprising two velocities and three stress components, pressure, pressure correction
and temperature. All of flow parameters are discretized using central differences, except
for the convective terms which are approximated by the linear–upwind differencing
scheme (LUDS) (Patankar and Spalding (1972)). This is the generalization of the well-
known up-wind differencing scheme (UDS), where the value of a convected variable at
a cell face location is given by its value at the first upstream cell center. In the linear-
upwind differencing scheme, the value of that convected variable at the same cell face is
given by a linear extrapolation based on the values of the variable at the two upstream
cells. It is, in general, the second-order accurate, as compared with first-order accuracy
of UDS, and thus, its use reduces the problem of numerical diffusion (Oliveira *et al.*
(1998)). The Cartesian reference coordinate system is located in the bottom boundary
and at left corner. Boundary conditions consist of two adiabatic walls in west and east
and two isothermal walls at north and south. For all boundaries, a no-slip condition is
imposed for the fluid velocity. The rest situation is used as the initial condition. The
used geometry and boundary conditions in this study are shown in Fig. 1. The geometry
has a rectangular shape with an aspect ratio of 2. Boundary conditions consist of two
isolated walls with zero gradient stress tensor components. The boundary conditions for



bottom and top plates are assumed a constant temperature so that the bottom plate has a
higher temperature. These boundaries have a zero gradient velocity and tensor
components, too.

**4. RESULTS AND DISCUSSION**
**4.1. Grid Study and Validation**
We perform some CFD simulations with different number of grids to study the
dependency of solution to mesh size. The meshes included quadratic elements. Table 2
lists the mean errors between average Nusselt number on horizontal lines on different
meshes and the $200 \times 100$ reference mesh. These errors are calculated for a viscoelastic
fluid with Giesekus model at $Ra = 227$. The numerical error decreases with increasing
the number of meshes as the mean error beings less than $0.08\%$ for mesh size greater
than $140 \times 70$. This finding indicates that a grid-independent solution is obtained when
using a mesh sizes larger than $140 \times 70$. To ensure that the obtained solution is grid-
independent, a mesh size of $150 \times 75$ was used for the CFD simulations.
As a benchmark comparison, simulations for free convection of Newtonian fluid
flow between two parallel plate have been carried out at $Ra = 10^4, 10^5, Pr = 100$. This
problem was studied previously by Khezar *et al.* (2012) and Turan *et al.* (2011) for
power-law fluid. The diagrams of average Nusselt number obtained from the present
study and work of Khezar *et al.* (2012) at *n*=1 are shown in Fig. 2a. As an additional
benchmark comparison, the distribution of dimensionless vertical velocity reported by



246 Turan *et al.* (2011) and the results obtained from the present study are illustrated in Fig.

247 2b at $Ra = 10^4 - 10^6$, $Pr = 100$ and $n = 1$. It is understood that in both cases, the results

248 of present CFD simulation have a suitable agreement with results of Khezar *et al.*

249 (2012) and Turan *et al.* (2011) with maximum error less than 3%.

250

251 **4.2. CFD Simulation of Mantle Convection Using Giesekus Model**

252 In this section, the effects of various parameters on flow regime of mantle convection

253 are studied. As observed in Eq. (4), the variation of parameters $\alpha$ and $\lambda$ could affect

254 the stress tensor field and this change in stresses will affect the velocity field.

255  According to the study of Pla *et al.* (2010), it could be inferred that with increasing

256 the exponential rate $\Gamma$, the circulations created by natural convection are moved toward

257 the bottom plate. It is resulted from the fact that by increasing $\Gamma$, the viscosity near

258 bottom plate would be decreased and the flow tends to circulate in this place. Also,

259 another parameter that effect on the flow and the circulation intensity is $\beta_G$. The results

260 of variations of these parameters will discuss in next sections. Remember that the

261 dependency of rheological and thermal properties and density on temperature and

262 pressure are considered and the variation of gravitational acceleration with depth of

263 Earth is modeled in following results.




Fig. 3 demonstrates a comparison between vertical velocity profiles of our
nonlinear viscoelastic model, power-law model (reported by Christensen (1983),
Cserepes (1982), Sherburn (2011), Van der Berg (1995), Yoshida (2012)) at n=3, and
the Newtonian model used by Pla *et al.* (2010). This Figure is presented in order to
compare the results of current CFD simulation (based on the non-linear Giesekus
consecutive equation, thermal-pressure dependence properties and depth dependence
gravitational acceleration) with previous simpler simulations that used Newtonian and
power-law models. As it is obvious, the velocity near upper plate for Giesekus model is
less than from the results of Pla *et al.* (2010) and power-law model. That is due to the
elastic force and higher value of viscosity at lower shear rates. Also, the maximum
vertical velocity of our simulation is smaller and the location of maximum vertical
velocity occurred upper than the location reported by Pla *et al.* (2010). That is because
of the viscoelastic portion of fluid behavior that we will discuss it in next sections. As it
is shown in Fig. 3, the depth in which the maximum velocity occurs is approximately
similar for power-law model and Giesekus constitutive equation. That is because of the
effect of apparent viscosity dependency to velocity gradient. Also noting to the velocity
profile, it is seen that all of models have the same results in vicinity of lower plate. But
for upper plate, the Figure demonstrates that the slope of vertical velocity for the
Giesekus model is smaller than the others. According to the Figure, there is a resistance
against the upward flow for Giesekus profile that two other models cannot predict it.
Actually, that is due to the consideration of elastic portion of fluid flow in our numerical
simulation. This finding indicated that the velocity and stress field have an obvious
deviation from Newtonian and generalized Newtonian behaviors by considering a non-





linear constitutive equation for mantle convection. In next sections, the effects of
material and thermal modules on mantle convection are studied based on the CFD
simulations that obtained using Giesekus non-linear model.

**4.2.1. Investigation of the Effect of Exponential Rate of Viscosity ($\Gamma$)**

We studied firstly the effect of increasing $\Gamma$ from zero to $10^{-3}$ on mantle convection.
This parameter represents the dependency of viscosity on temperature variation. Fig. 4
shows the streamlines for different values of $\Gamma$ at $\beta_G = 0.98$, $\alpha = 0.2$ and
$En = 6.04 \times 10^{32}$. It is evident from Fig.4 that the circulations in the mantle physically
depend on $\Gamma$. As the exponential rate ($\Gamma$) is increased, the maximum velocity in
geometry is enhanced and the circulations moved downward. According to Eq. (6), the
dependency of viscosity of mantle on temperature is more increased by enhancing the
exponential rate ($\Gamma$). In other words, by increasing the exponential rate ($\Gamma$), the
viscosity is more decreased near to the lower plate (high temperature region) and the
fluency of mantle is intensified. Therefore, it is expected that the velocity of mantle
convection is enhanced by increasing the exponential rate. The results show that an
increment of 1.6% in vertical velocities by increasing the exponential rate from zero to
$10^{-5}$, 17.1% growth by increasing $\Gamma$ to $10^{-4}$ and with enhancing the $\Gamma$ from zero to
$10^{-3}$ it growths up to 4.32 times. The CFD simulations indicated that the effect of
exponential rate on maximum value of velocity is nonlinear. The contours of axial
normal stress and shear stress are shown in Fig. 5. As it is obvious, the exponential rate




has a significant influence on magnitude of stress fields that is increased by enhancing
the exponential rate. As an example, for $\Gamma = 10^{-4}$, the value of dimensionless stress
component $\tau_{xx}$ becomes 1.1 times greater than the one with exponential rate of zero.
Also, with increasing the value of $\Gamma$ by $10^{-3}$, it growths up to 2.56 times. Actually, with
increasing the exponential rate, the dependency of viscosity on temperature is
intensified and then the right hand side of Eq. (4) increases so this change leads to
enhancement of stress field. Fig. 6 displays the location of maximum vertical velocity at
$Y/H = 0.5$ versus the exponential rate. The dimensionless depth of points $Y$, where the
maximum of velocity is occurred, is $Y = 0.5$ for $\Gamma = 0$ and by increasing the
exponential rate to $10^{-5}$, this depth will be decreased to 2.4%. The amount of this
reduction for $\Gamma = 10^{-4}$ and $\Gamma = 10^{-3}$ is 10% and 24%, respectively. We obtained the
following relation for location of maximum vertical velocity with 95% confidence:

$$Y = -10.58\Gamma^{\frac{2}{3}} + 0.4933$$

The above correlation is used in plotting the Fig. 6. The downward movement of
location of maximum vertical velocity with increasing exponential rate could be
attributed to shifting the center of vortices which is shown previously in Fig. 5.
In Fig. 7, the temperature distribution in mantle is shown. According to this Figure,
heat transfer regime is almost conduction. Nevertheless, closer looking to the
temperature distribution, some convection behavior could be observed. The temperature
profile on a horizontal line is shown in Fig. 8. As it is expected, the temperature profile
shown in Fig. 8 has a minimum value at mid of horizontal line and the maximum values



are located at left and right hand sides of numerical domain. Fig. 9 shows the stress
magnitude on upper plate for different value of $\Gamma$ at $\alpha = 0.2$ and $\lambda = 1.5 \times 10^{13}\, s$. As
expected from Eq. (6), the viscosity will be more depended on temperature by
increasing the value of $\Gamma$. Thus, the viscosity will be decreased with increasing $\Gamma$ and
in the other hand; the velocity field will be intensified that the participation of these
factors determines stresses in vicinity of upper plate. According to Fig. 9, in the case of
$\Gamma = 10^{-5}$, with increasing $\beta_G$ from 0.5 to 0.8, the maximum stress magnitude is
increased by 32.2% and by enhancing $\beta_G$ to 0.9 and 0.98, the growing percentages are
32.2% and 101%, respectively. As mentioned before, there are several factors that affect
the flow pattern such as $\Gamma$ and $\beta_G$. The result of this participation clearly is seen here,
when the viscosity ratio vary from 0.9 to 0.98, it seems that in this interval, the effect of
these two parameters ($\Gamma$ and $\beta_G$) is neutralized each other and lead to having the same
stress magnitude at these points.

**4.2.2. Investigation of the Effect of Viscosity Ratio ($\beta_G$)**
The parameter $\beta_G$ is a criterion portion for demonstration of domination of viscoelastic
towards pure Newtonian portions of fluid behavior. In fact, when this parameter is much
closer to unity, the viscoelastic behavior is dominated and when $\beta_G$ is close to zero, the
pure Newtonian behavior of fluid is dominated. As it is shown in Fig. 10, by increasing
$\beta_G$ from 0.8 to 0.98, the stress magnitude on upper plate has been increased, but the





vertical velocity near to the both lower and upper plates is decreased. This effect is
related to the higher value of viscosity of viscoelastic potion in comparison of pure
Newtonian behavior that causes increasing the total viscosity and decreasing the fluidity
of model (refer to Eq. 3). This finding is approved by the data of maximum magnitude
of shear stress near to the upper plate which is reported in Table 3. According to the
Table, $\tau_{max}$ is increased by enhancing the viscosity ratio which is caused from
increasing the fluid viscosity.

356        Fig. 11 shows variation of normalized vertical velocity on a vertical line for

different values of exponential rates ($\Gamma$) and viscosity ratios ($\beta_G$). As it is understood
from Fig. 11, in constant viscosity ratio, when $\Gamma$ is increased, the velocities are
increasing very strongly, but as viscosity ratio changes, a contrast occurred between
these two factors (as it is shown in Fig. 11c, the velocities are increased and in Fig. 11b,
the vertical velocities are decreased). In other word, at $\beta_G = 0.9$, the effect of exponential
rate is prevailed but with increasing the viscosity ratio to $\beta_G = 0.98$, the effect of
viscosity ratio is dominated.

**4.2.3. Investigation of the Effect of Elasticity**
The elastic number is generally used to study the elastic effect on the flow of
viscoelastic fluids. According to the Eq. 9, the elastic number is defined as the ratio of
Weissenberg to Reynolds numbers. This dimensionless group is independent from
kinematic of flow field and it is only depended on material modules for a given



geometry. Here, the elastic number is proportional with relaxation time of model and it
is increased by enhancing the material elasticity. Figs. 12 and 13 display velocity and
stress magnitude for different values of elastic number. Table 4 presents the value of
maximum normalized vertical velocity for different elastic numbers and various
viscosity ratios. According to the Fig. 12, the velocity of mantle convection is decreased
by increasing the elastic number from $6.04 \times 10^{26}$ to $6.04 \times 10^{32}$ and it is increased by
increasing the elastic number to $6.04 \times 10^{32}$. The first decreasing in the normalized
velocity could be attributed to increasing the normal stresses resulted from fluid
elasticity. In the other word, some main portion of energy of convection is stored as the
elastic normal stresses. In larger elastic numbers, the effective viscosity of flow is
decreased which is related to the nature of nonlinear dependency of viscometric
function of Giesekus constitutive equation on relaxation time at large enough elastic
numbers (Bird *et al.* (1987)).

**4.2.4. Investigation of Mobility Factor Effect**
Fig. 14 shows the effects of mobility factor on the vertical velocity for different values
of viscosity ratio.  Due to the non-linear nature of our viscoelastic model and the high
elastic number, anticipation of effects of all factors is not easy and it is strongly affected
by the variation of other factors. Regarding to high viscosity of mantle, the effect of
mobility factor must be minimal, as it is shown in Fig. 14. The effects of mobility factor
are only important near both upper and lower plate. In the other word, the main
variation of velocity distributions with changing the mobility factor occurs in the upper



and lower plate. For $\alpha = 0.05$, the magnitudes of normalized velocities in vicinity of
upper plate are increasing by enhancing $\beta_G$ from 0.5 to 0.9 between 20% to 50% and
with increasing the viscosity ratio to 0.98, the velocities are decreasing about 70%. In
contrast, for the lower plate, this variation is reversing, *i.e.,* the velocities with
increasing $\beta_G$ to 0.9 are decreasing. The same effect is available for $\alpha = 0.2$. Also, the
variation of velocity near upper plate for $\alpha = 0.1$ and 0.3 are similar. In these cases, with
increasing $\beta_G$ from 0.5 to 0.9, the velocities in this place are decreasing and with
increasing the viscosity ratio to 0.98, the magnitudes of velocities are ascending. Table
5 presents the maximum normalized vertical velocity for various values of elastic
numbers and different viscosity ratios.

**4.2.5. Investigation of the Effect of Rayleigh Number**
If we want to study natural convection and investigate the strength of convection, the
Rayleigh number is a suitable criterion for this aim. Since mantle convection has a low
Rayleigh number, thus the temperature field should have a conductive form (see Fig 7).
According to Eq. (10), the Rayleigh number is a function of temperature, so it is varying
all over the geometry because the viscosity is temperature dependent and is varying.
Fig.15 presents the streamlines for different Rayleigh numbers. According to Fig. 15, by
increasing the Rayleigh number, the velocity in geometry is increased and the
circulations move downward and get more intense. By increasing *Ra* from 22.7 to 227,
the velocity magnitude will vary with order of $10^1$. If we rise the Rayleigh number to



1135, this growth in velocities is in order of $10^2$ and when we set the *Ra* as 2270, the
velocity magnitude will be in order of $10^3$. It is important to remember that the
temperature difference between the hot and cold plates is the potential of mantle
convection so the velocity is increased by increasing the Rayleigh number. Fig. 16
shows the stress contours for various Rayleigh number. The Figure shows that with
increasing the Rayleigh number, the maximum stress in geometry has enhanced
significantly. This effect is related to increasing the shear rate of flow field which is
intensifying the stress field. According to the Figure, the Giesekus model predicts a
large shear stress in comparison of normal stress components which is related to the
shear flow behavior of mantle convection which has a suitable agreement with previous
reports that used other constitutive equations (Ghias and Jarvis (2008), Severin and
Herwig (1999), Pla *et al.* (2009), Hirayama and Takaki (1993), Fröhlich *et al.* (1992),
Tomohiko *et al.* (2004)).

**5. CONCLUSIONS**

428        Current study deals with a numerical simulation of mantle convection using a

temperature dependent nonlinear viscoelastic constitutive equation. The effect of
temperature on rheological properties consisting of the viscosity, normal stress
differences and relaxation time of mantle are modeled using appropriate equations of
state which were the main innovative aspects of current study. The variation of
gravitational acceleration with depth of Earth and the effect of the work of stress field





(viscous dissipation) on mantle convection were simulated for the first time. According
to the literature, the previous studies were restricted to the linear and quasi-linear
viscoelastic constitutive equations and the nonlinearity nature of mantle convection was
modeled using simple nonlinear constitutive equations just for apparent viscosity such
as the power-law and cross models. The Giesekus nonlinear viscoelastic model was
used as the constitutive equation in present study. This high order nonlinear model was
used because of large-scale creeping viscoelastic flow of mantle convection in space
and time. Using Giesekus constitutive equation, we present a more accurate solution for
this problem because of taking into account of shear-dependent nonlinear viscometric
functions, the effects of third invariant of shear rate tensor on stress field, and effects of
material elasticity for large deformations of mantle.
It is important to remember that the non-linear constitutive equations such as the
Giesekus equation could able to model the material elasticity and relaxation spectra
much better than linear models for large deformations of flow field. We also showed
that the result of this model has an obvious deviation from pure Newtonian and power-
law solutions that reported in literatures.
The effect of temperature on viscosity of the mantle is studied, firstly. The results
show that increasing of exponential viscosity rate led to the enhancing the maximum
velocity and making the circulation moving downward so that with increasing $\Gamma$ from
zero to $10^{-3}$, an increase of 4.32 times in vertical velocity and an increase of 2.56 times
in $\tau_{xx}$ were obtained. A formula have presented for the position of maximum vertical
velocity as a function of $\Gamma$. The effect of viscosity ratio is also investigated on the



mantle convection. These results not only show how stress magnitude on upper plate
increases by enhancing the viscosity ratio from 0.8 to 0.98, but also prove decreasing of
the vertical velocity near to the both lower and upper plates. These effects are related to
the higher value of viscosity of viscoelastic Gesikus model relative to the pure viscous
portion (Newtonian behavior) which causes decreasing of fluidity of mantle convection.
In constant viscosity ratio, when $\beta_G$ increases, the velocities are rising very strongly,
but as viscosity ratio changes, a competition occurred between these two factors. In
other word, at $\beta_G = 0.9$, the effect of exponential rate is prevailed but with increasing
the viscosity ratio up to $\beta_G = 0.98$ the effect of viscosity ratio is dominated and the
velocities are descended. The variation of Elastic number shows the nature of nonlinear
dependency of viscometric function of Giesekus constitutive equations on relaxation
time at large enough elastic numbers. Present study indicates decreasing of effective
viscosity flow for larger elastic numbers. The obtained results show how main
variations of velocity distributions with changing of mobility factor occur in the upper
and lower plates. Here, the effect of Rayleigh number on mantle convection is also
investigated and characterized that with increasing the Rayleigh number, the maximum
stress in geometry has enhanced significantly. This effect is related to increasing the
shear rate of flow field which is intensifying the stress field.
Future works could be focused on the effect of mantle convection on plate motions,
effect of chemical reactions occurring in the mantle, and plumes growing by
considering a non-linear viscoelastic consecutive equation.





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

570      1063.








**Table 1.** Parameters related to mantle convection (Pla *et al.* (2010)).

| Parameter | Value |
|---|---|
| $H\,[m]$ | $2.9 \times 10^{6}$ |
| $\kappa\,[m^{2}s^{-1}]$ | $7 \times 10^{-7}$ |
| $\beta_{T}\,[K^{-1}]$ | $10^{-5}$ |
| $\nu\,[m^{2}s^{-1}]$ | $3.22 \times 10^{20}$ |
| $Pr$ | $10^{26}$ |
| $Ra$ | $3.48\,\Delta T$ |









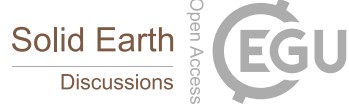



**Table 2.** Percentage of mean absolute errors between average velocity obtained from

different meshes and the $200 \times 100$ reference mesh.

| Ra | $N_x \times N_y$ | | | | |
|---|---|---|---|---|---|
| | $100 \times 50$ | $120 \times 60$ | $140 \times 70$ | $150 \times 75$ | $170 \times 85$ |
| 227 | 0.1858 | 0.1283 | 0.0812 | 0.0602 | 0.0314 |












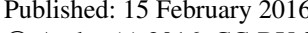





**Table 3.** Maximum magnitude of stress on top plate for different values of $\beta_G$ and $\Gamma$

($\alpha = 0.2$ and $En = 6.04 \times 10^{32}$).

| $\beta_G$ | $\tau_{max}$ | | | |
|---|---|---|---|---|
| | $\Gamma = 0$ | $\Gamma = 10^{-5}$ | $\Gamma = 10^{-4}$ | $\Gamma = 10^{-3}$ |
| 0.98 | 36.8 | 37 | 40.5 | 133.75 |
| 0.9 | 30.6 | 33.75 | 32.6 | 112.5 |
| 0.8 | 29.5 | 29.8 | 32.6 | 112.5 |
| 0.5 | 18.25 | 18.4 | 20.1 | 73 |













**Table 4.** Maximum magnitude of vertical velocity on a vertical line at $x$=1 for different

values of $\beta_G$ and $En$ ($\alpha = 0.2$ and $\Gamma = 10^{-5}$).

| $\beta_G$ | $V_{max}$ | | | | | |
|---|---|---|---|---|---|---|
| | $En = 6.04 \times 10^{26}$ | $En = 6.04 \times 10^{28}$ | $En = 6.04 \times 10^{30}$ | $En = 6.04 \times 10^{32}$ | $En = 6.04 \times 10^{34}$ | $En = 6.04 \times 10^{36}$ |
| 0.50 | 0.0400 | 0.0410 | 0.0390 | 0.0392 | 0.0396 | 0.0395 |
| 0.80 | 0.0387 | 0.0400 | 0.0395 | 0.0439 | 0.0361 | 0.0400 |
| 0.90 | 0.0427 | 0.0380 | 0.0390 | 0.0385 | 0.0380 | 0.0410 |
| 0.98 | 0.0359 | 0.0423 | 0.0420 | 0.0341 | 0.0410 | 0.0373 |












**Table 5.** Maximum magnitude of vertical velocity on a vertical line at x=1 for different

values of $\beta_G$ and $\alpha$ ( $En = 6.04 \times 10^{32}$ and $\Gamma = 10^{-5}$ )

| $\beta_G$ | $V_{max}$ | | | | | |
|---|---|---|---|---|---|---|
| | $\alpha = 0.05$ | $\alpha = 0.10$ | $\alpha = 0.20$ | $\alpha = 0.30$ | $\alpha = 0.40$ | $\alpha = 0.50$ |
| 0.50 | 0.0395 | 0.0397 | 0.0397 | 0.0398 | 0.0397 | 0.0395 |
| 0.80 | 0.0398 | 0.0356 | 0.0439 | 0.0407 | 0.0407 | 0.0385 |
| 0.90 | 0.0376 | 0.0390 | 0.0385 | 0.0380 | 0.0417 | 0.0424 |
| 0.98 | 0.0385 | 0.0383 | 0.0341 | 0.0385 | 0.0415 | 0.0373 |



