# Peer review of "Numerical simulation of mantle convection using a"

_Solid Earth, 2016_

## Short Comment (SC1) · 16 Feb 2016

First of all, the copy I was able to download has no figures, which makes it difficult to assess the results in more detail. The authors mention that their inclusion of the depth dependence of gravity is a special feature of their study, but I do not see any substantial discussion of what is gained through it. It is true that in the case of Earth, depth dependence of g is not usually considered, simply because g is almost constant through the mantle. I am perplexed that the authors derive their depth dependence from a paper from Bullen (1939) and do not even mention PREM (Dziewonski & Anderson, PEPI, 1981), which is probably still the default reference to use if one doesn't derive the gravity profile self-consistently from the model. If I plot the eq. 13 they derived from Bullen, it turns out that it is almost constant down to ca. 2500 km depth but deviates strongly from PREM in the lowermost mantle; it yields a g value of 15 m/sˆ2 at the CMB, while
[Figure]

PREM hardly exceeds 10.5. I would expect this to promote instabilities rising from the CMB much more easily than it should be according to PREM, and I wonder what this effect does to their models.

---

## Referee Comment (RC1) · Anonymous Referee #1 · 22 Feb 2016

I previously reviewed this paper for another journal. As this manuscript seems identical (with no attempt to incorporate the recommendations I made previously) I will simply repeat what I wrote then:

Here the authors present some convection simulations using a nonlinear viscoelastic rheology with the Giesekus constitutive equation, and claim that they are relevant to understanding Earth's mantle. There are several reasons not to publish this manuscript in Solid Earth.

Most fundamentally: Does the Giesekus constitutive relationship that they use actually apply to rocks? This relationship was derived specifically for polymers. Rocks are not polymers – they are crystalline. Are there any laboratory experiments showing that rock minerals, or indeed any type of crystalline material, has nonlinear viscoelasticity

that matches the Giesekus constitute relationship? Or any field observations? Unless the authors can cite some evidence that this is actually relevant, then this manuscipt has no place in an Earth science journal.

Secondly, in a journal like SE one expects a strong link between the presented results and the actual Earth, but instead the authors present a nondimensional parameter study with no attempt to make quantitative inferences or predictions regarding the actual Earth. This is in contrast to earlier studies on mantle convection with viscoelasticity, which the authors don't seem to know about since they don't cite them (Beuchert and Podladchikov, 2010; Harder, 1991; Moresi et al., 2003; Moresi et al., 2002) (also there are many papers on lithosphere & crust deformation that include viscoelasticity). The usual view is that in the mantle elasticity is unimportant at geological time scales because the viscoelastic relaxation time is short compared to the deformation time scale (viscosity/shear modulus $\sim$1e21/1e11 $\sim$ 300 years), while in the lithosphere, viscosity can be orders of magnitude higher so elasticity can be important. It is not clear that the authors reach this regime. As it is, this paper belongs more in a journal like Journal of Non-Newtonian Fluid Mechanics.

Thirdly, the investigative method. If the authors want to demonstrate that nonlinear viscoelasticity is important then they need to show corresponding solutions for (i) nonlinear viscoelasticity (ii) linear viscoelasticity and (iii) viscous flow, and identify how and where they are different.

Fourthly, they claim that they include viscous dissipation "for the first time" but in fact there are several 10s of papers on mantle convection that include this term – basically anything study that uses a compressible approximation. Here are just a few examples to get them started: (Balachandar et al., 1993; Glatzmaier, 1988; Jarvis and McKenzie, 1980; Leng and Zhong, 2008; Tackley, 1996).

Fifthly: they claim that including variation of "g" through the mantle is an important new aspect of their study: actually the variation with depth is very small – only a few

% - as you can see in the widely-used PREM (Preliminary Reference Earth Model) (e.g. http://geophysics.ou.edu/solid_earth/prem.html). This is why it is almost always ignored.

I could go on, but that's enough for now.

REFERENCES

Balachandar, S., Yuen, D.A., Reuteler, D., 1993. Viscous and Adiabatic Heating Effects In 3-Dimensional Compressible Convection At Infinite Prandtl Number. Physics Of Fluids a-Fluid Dynamics 5, 2938-2945.

Beuchert, M.J., Podladchikov, Y.Y., 2010. Viscoelastic mantle convection and lithospheric stresses. Geophys. J. Int. (UK) 183, 35-63.

Glatzmaier, G.A., 1988. Numerical simulations of mantle convection - time-dependent, 3-dimensional, compressible, spherical-shell. Geophys. Astrophys. Fluid Dyn. 43, 223-264.

Harder, H., 1991. Numerical-simulation of thermal-convection with Maxwellian viscoelasticity. Journal Of Non Newtonian Fluid Mechanics 39, 67-88.

Jarvis, G.T., McKenzie, D.P., 1980. Convection in a compressible fluid with infinite Prandtl number. J. Fluid Mech. 96, 515-583.

Leng, W., Zhong, S., 2008. Viscous heating, adiabatic heating and energetic consistency in compressible mantle convection. Geophys. J. Int. 173, 693-702.

Moresi, L., Dufour, F., Muehlhaus, H.-B., 2003. A Lagrangian integration point finite element method for large deformation modeling of viscoelastic geomaterials. J. Comp. Phys. 184, 476-497.

Moresi, L., Dufour, F., Muhlhaus, H., 2002. Mantle convection models with viscoelastic/brittle lithosphere: Numerical methodology and plate tectonic modeling. Pure and Applied Geophysics 159, 2335-2358.

Tackley, P.J., 1996. Effects of strongly variable viscosity on three-dimensional compressible convection in planetary mantles. J. Geophys. Res. 101, 3311-3332.

---

## Referee Comment (RC2) · Anonymous Referee #2 · 22 Feb 2016

I have a number of serious issues with this work, as summarized below.

To start with, the pdf does not have figures which makes it hard to review it. Apart from this sloppiness, there are a large number of other issues that I believe are serious and as such this is not publishable in its current form.

1) Lack of literature review The authors give a very rather incomplete review of previous work on mantle convection, for which an enormous amount of literature exists (interestingly they miss seminal work by McKenzie, Christensen, Tackley, Moresi, Solomatov and many others). I strongly suggest that they read up on the topic, for example by reading textbooks by Schubert, Davies, or some of the many review papers (by Tackley, Bercovici, Ricard, many of which are available online).

As the current paper deals with viscoelastic convection, I had expected at least a complete review of existing work on this topic which is however also not the case, which shows a rather large ignorance towards previous work.

Thielmann, M., Kaus, B., and Popov, A.A., 2015, Lithospheric stresses in Rayleigh–Bénard convection: effects of a free surface and a viscoelastic Maxwell rheology: Geophysical Journal International, v. 203, no. 3, p. 2200–2219, doi: 10.1093/gji/ggv436.

Beuchert, M.J., and Podladchikov, Y.Y., 2010, Viscoelastic mantle convection and lithospheric stresses: Geophysical Journal International, v. 183, p. 35–63

Muhlhaus, H.-B., Davies, M., and Moresi, L., 2006, Elasticity, Yielding and Episodicity in Simple Models of Mantle Convection: Pure and Applied Geophysics, v. 163, no. 9, p. 2031–2047, doi: 10.1007/s00024-006-0111-5.

Muhlhaus, H.-B., and Regenauer-Lieb, K., 2005, Towards a self-consistent plate mantle model that includes elasticity: simple benchmarks and application to basic modes of convection: Geophysical Journal International, v. 163, no. 2, p. 788–800..

Moresi, L., Dufour, F., and Muehlhaus, H.B.M.X., 2002, Mantle Convection Modeling with Viscoelastic/Brittle Lithosphere: Numerical Methodology and Plate Tectonic Modeling: Pure and Applied Geophysics, v. 159, no. 10, p. 2335–2356, doi: 10.1007/s00024-002-8738-3.

and this should certainly include the pioneering work of Harder in this respect: Harder, H., 1991, Numerical-Simulation of Thermal-Convection with Maxwellian Viscoelasticity: Journal of Non-Newtonian Fluid Mechanics, v. 39, no. 1, p. 67–88.

2) Gravity Some of the other reviewers were a bit annoyed of your use of a depth-varying g, yet your polynomial does indeed reproduce the depth dependent effect of g on Earth, if I plot it on MATLAB, using the following lines:

» y=[1:3000]/1000; » g=-0.118*y.^6+ 0.602*y.^5 - 1.006*y.^4 + 0.6884*y.^3 - 0.3708*y.^2 + 0.167.*y - 9.846; » plot(g,y*1000), axis ij

The point is, however, that the variation of g within the Earth's mantle is very minor which your expression also shows, which is why it is usually assumed to be constant.

3) Employed constitutive relationships All above mentioned papers employ linear viscoelasticity, for the simple reason that there is not much data to support the use of more complicated elasticity models for applications on the scale of a convecting mantle (apart maybe from using a Kelvin body for bulk deformation). One can ofcourse come up with arbitrary complex constitutive relationships but if there is no data to back it up you are not modelling a problem that is geoscientifically relevant. It is unclear to me why the Giesekus model should be relevant for geoscientific applications and you don't give a justification for that which implies that your paper is simply irrelevant for geoscientific purposes.

4) Viscous dissipation Different than what you claim, viscous dissipation has been studied in the context of mantle convection models since at least the mid-80ies (please look at classical papers by Christensen and the above mentioned textbooks). Importantly, you should include adiabatic heating if you include viscous dissipation as they are of similar order of magnitude.You don't do that here.

5) Inertial terms Mantle convection is low Reynolds number fluid flow; yet you include inertial terms in your formulation. You can use that to model mante convection but you would have to employ a very small timestep; moreover, you should first demonstrate that your code actually reproduces normal viscous convection for example by reproducing the Blankenbach benchmarks.

6) Missing figures the uploaded pdf has no figure which makes it impossible to review it. Yet, from the text alone it is clear that this work is currently very far removed from being publishable in an Earth Science journal.

---

## Author Comment (AC1) · 13 May 2016

**To the T. Ruedas**

Many thanks for the referee's valuable comments and his/her time spent in reviewing our paper (SE-2016-12). I would like to mention that in the revised paper, all the points have been taken into consideration. Following, I refer the comments made by the referee with their corresponding answers as italic font. The changes are highlighted in the revised paper (in yellow color).

1. First of all, the copy I was able to download has no figures, which makes it difficult to assess the results in more detail. The authors mention that their inclusion of the depth dependence of gravity is a special feature of their study, but I do not see any substantial discussion of what is gained through it. It is true that in the case of Earth, depth dependence of g is not usually considered, simply because g is almost constant through the mantle. I am perplexed that the authors derive their depth dependence from a paper from Bullen (1939) and do not even mention PREM (Dziewonski & Anderson, PEPI, 1981), which is probably still the default reference to use if one doesn't derive the gravity profile self-consistently from the model. If I plot the eq. 13 they derived from Bullen, it turns out that it is almost constant down to ca. 2500 km depth but deviates strongly from PREM in the lowermost mantle; it yields a g value of 15 m/sˆ2 at the CMB, while PREM hardly exceeds 10.5. I would expect this to promote instabilities rising from the CMB much more easily than it should be according to PREM, and I wonder what this effect does to their models.

**Response:** *We uploaded the Figures' file in the system, but unfortunately, figures were not included in the final online file of our article.*

*About the reference used for variation of gravitational acceleration in mantle (i.e. Bullen, 1939), we should say that there is a maximum difference of 1.8% between the data provided by Bullen (1939) and Dziewonski and Anderson (1981). Thus, using gravitational acceleration data of each reference would lead to similar results. However, in order to inform the readers of our paper about the newer data of this parameter, we add this reference in the final version of our paper (refer to page 13 of the revised paper):*

> *... Also, one can refer to the gravitational acceleration data of Dziewonski and Anderson (1981) which provide a comprehensive data set on the variation of gravitational acceleration in mantle.*

The variable $y = \tilde{y} / H$ in Equation (13) is dimensionless. Thus, it should be evaluated in the range of 0 to 1. If the values of Eq. (13) are calculated in the range of 0 to 1, one can see that the maximum of gravitational acceleration will be around $10.14\,\mathrm{ms}^{-2}$. Thus, it should not cause any instability in the solution procedure.

Best,

M. Norouzi

---

## Author Comment (AC2) · 13 May 2016

**To the Reviewer #1**

Many thanks for the referee's valuable comments and his/her time spent in reviewing our paper (SE-2016-12). I would like to mention that in the revised paper, all the points have been taken into consideration. Following, I refer the comments made by the referee with their corresponding answers as italic font. The changes are highlighted in the revised paper (in yellow color).

**Comment#1**: Does the Giesekus constitutive relationship that they use actually apply to rocks? This relationship was derived specifically for polymers. Rocks are not polymers – they are crystalline. Are there any laboratory experiments showing that rock minerals, or indeed any type of crystalline material, has nonlinear viscoelasticity that matches the Giesekus constitute relationship? Or any field observations? Unless the authors can cite some evidence that this is actually relevant, then this manuscipt has no place in an Earth science journal.

**Response**: *It is pleasure for me that it is prepared an opportunity for us to clarify some uncertainties. The main comment of respectable reviewer is: Just linear viscoelastic models (such as Maxwell model) are suitable for modeling the mantle convection and Giesekus constitutive equation is not a good choice due to difference of the class of materials. I should present some clarifications that should be useful for readers. Since 1950, a movement in rheology is begun to present constitutive equations for viscoelastic materials especially the polymeric materials. Not only the Giesekus constitutive equation but also all of famous linear and nonlinear constitutive equations (such as **Maxwell model**, power-law equation, cross equation and ...) have been presented for polymers. These models have been used in other branches of science for solving the flow and deformation of viscoelastic materials such as geology, biotechnology, soil engineering, chemical engineering, food engineering and so on. For example, the power-law model is a simple nonlinear constitutive equation that can be model the nonlinear shear dependent viscosity using the second invariant of shear rate tensor to define the generalized shear rate. This model is widely used for solving the flow of non-Newtonian liquids due to its simplicity. **The power-law model was also used for modeling the mantle convection**. A summary for the type of models that used in previous studies is presented in following Tables (tables located after the response to this comment). In these Tables, **6 works** are listed that used the power-law model as the constitutive equations.*

*The respectable reviewer should notice to this problem: why did the previous researchers use the **power-law** equation for **modeling the mantle convection** (as a too simple nonlinear model which is basically presented for **polymeric liquids**)? The answer is: Not only the class of material but also the **type of deformation** is important in selecting a constitutive equation for any rheological problems. The previous researchers used the power-law model to solve the mantle convection due **to large scale deformation** (The flow is a large scale deformation). A same approach has been performed to solve the mantle convection problem using the linear viscoelastic model to study the effect of material **elasticity** on the problem but the result of these models are not proper for large scale deformation. The main motivation of present study is answering to this question: "is it possible to study the both effect of material elasticity and nonlinear viscosity on mantle convection?" The answer is using the nonlinear viscoelastic constitutive equations such as Giesekus model. The model can present simultaneously a fractional **nonlinear viscosity** (similar to power-law model) and **elasticity** (similar to linear differential viscoelastic models). Because of using the convective coordinate system in its definition (using complicated upper convected derivations instead of simple time derivations), it is so suitable for modeling the large scale deformations that is the main advantage of this model on linear models. In other word, the model is able to keep the memory of deformation like as linear integral viscoelastic models. It is important to mention that the Giesekus model can be simplified to **linear Jeffries model** for small deformations. Therefore, the authors believe that this complicated nonlinear constitutive equation can better model the mantle convection (as a nonlinear-viscoelastic-large scale flow) by selecting the suitable constants of this model (viscosity, relaxation time and mobility factor). This finding was also mentioned by Prof. Harder (1991) in conclusion of his study about the modeling of mantle convection as a suggestion for future studies:*

> *"This study has demonstrated the major differences between convection with Maxwellian versus Newtonian rheology and shown that thermal convection is a very suitable test case for numerical methods simulating viscoelastic flow. It has been possible to extend the simulation up to Deborah numbers De = 1.0, which is sufficient to induce significant changes in the flow fields. A main new feature at high De is the presence of a normal stress singularity along the boundary, which is absent in Newtonian or in low-De flow. Since this singularity is starting from the stagnation points at the cell corners, this behaviour is presumably related to the well known singularity of the upper convected Maxwell model in a pure shear*

*flow [4,5]. It is common experience [4,7,13] that rheological models with more realistic response are numerically easier to handle. Examples are the Jeffreys model and its non-linear generalisations, i.e. the **Giesekus** and Leonov models (Harder (1991))".*

*The work of Harder (1991) is reported in the revised manuscript and some explanations about selecting the nonlinear Giesekus model is inserted to the revised paper (refer to pages 4, 5 and 7).*

| Constitutive model | References |
|---|---|
| **Maxwell** | [*]OzBench, M., Regenauer-lieb, K., Stegman, D.R., Morra, G., Farrington, R., Hale, A., May, D.A., Freeman, J., Bourgouin, L., Muhlhaus, H., Moresi, L., 2008. A model comparison study of large-scale mantle-lithosphere dynamics driven by subduction. Phys. Earth Planet. Int. 171, 224–234. |
| | [*]Thielmann, M., Kaus, B.J.P., Popov, A.A., 2015. Lithospheric stresses in Rayleigh– Bénard convection: effects of a free surface and a viscoelastic Maxwell rheology. Geophys. J. Int., 203, 2200–2219. |
| | [*]Harder, H., 1991. Numerical simulation of thermal convection with Maxwellian viscoelasticity. Journal of Non-Newtonian Fluid Mechanics, 39, 67–88. |
| | [*]Moresi, L., Dufour, F., Muhlhaus, H.B., 2002. Mantle Convection Modeling with Viscoelastic/Brittle Lithosphere: Numerical Methodology and Plate Tectonic Modeling. Pure Appl. Geophys. 159, 2335–2356. |

[*]The asterisk sign means that the reference has been mentioned in our paper.

| Constitutive model | References |
|---|---|
| **Power-law** | King, S.D., Gable, C.W., Weinstein, S.A., 1992. Models of convection-driven tectonic plates: a comparison of methods and results. Geophys. J. Int., 109, 481-487. |
| | [*] Gerya, T.V., Yuen, D.A., 2007. Robust characteristics method for modelling multiphase visco-elasto-plastic thermo-mechanical problems. Phys. Earth Planet. Int. 163, 83–105. |
| | [*] Christensen, U., 1983. Convection in a variable-viscosity fluid: Newtonian versus power-law rheology. Earth and Planetary Science Letters, 64, 153–162. |
| | [*] Cserepes, L., 1982. Numerical studies of non-Newtonian mantle convection. Physics of the Earth and Planetary Interiors, 30, 49–61 |
| | [*] Van den Berg, Arie P., Yuen, D.A., Van Keken P.E., 1995. Rheological transition in mantle convection with a composite temperature-dependent, non-Newtonian and Newtonian rheology. Earth and Planetary Science Letters, 129, 249–260. |
| | Gerya, T.V., Yuen, D.A., 2003. Characteristics-based marker-in-cell method with conservative finite-differences schemes for modeling geological flows with strongly variable transport properties. Physics of the Earth and Planetary Interiors, 140, 293–318. |
| | Muhlhaus, H.S., Regenauer-Lieb, K., 2005. Towardsa self-consistent plate mantle model that includes elasticity: simple benchmarks and application to basic modes of convection. Geophys. J. Int., 163, 788–800. |

[*] The asterisk sign means that the reference has been mentioned in our paper.

| Constitutive model | References |
|---|---|
| **Newtonian, temperature- or pressure-dependent** | [*] Yanagawa, T.K.B., Nakada, M., Yuen, D.A., 2004. A simplified mantle convection model for thermal conductivity stratification. Phys. Earth Planet. Int. 146, 163–177. |
| | Stemmer, K., Harder, H., Hansen, U., 2006. A new method to simulate convection with strongly temperature-and pressure-dependent viscosity in a spherical shell: Applications to the Earth's mantle |
| | [*] Moresi, L.N., Solomatov, V.S., 1995. Numerical investigation of 2D convection with extremely large viscosity variations. Phys. Fluids 7, 2154–2162. |
| | [*] Pla, F., Herreroa, H., Lafitte, O., 2010. Theoretical and numerical study of a thermal convection problem with temperature-dependent viscosity in an infinite layer. Physica D 239, 1108–1119. |
| | [*] Hansen, U., Yuen, D.A., Kroening, S.E., Larsen, T.B., 1993. Dynamical consequences of depth-dependent thermal expansivity and viscosity on mantle circulations and thermal structure. Physics of the Earth and Planetary Interiors, 77, 205–223 |
| | Kronbichler, M., Heister, T., Bangerth, W., 2012. High accuracy mantle convection simulation through modern numerical methods. Geophys. J. Int., 191, 12–29. |
| | [*] Kameyama, M., Kageyama, A., Sato, T., 2008. Muligrid-based simulation code for mantle convection in spherical shell using Yin-Yang grid. Phys. Earth Planet. Int. 171, 19–32. |
| | Yoshida, M., 2010. Preliminary three-dimensional model of mantle convection with deformable, mobile continental lithosphere. Earth and Planetary Science Letters, 295, 205–218 |
| | [*] Kellogg, L.H., King, S.D., 1997. The effect of temperature dependent viscosity on the structure of new plumes in the mantle: results of a finite element model in a spherical, axymmetric shell. Earth Planet. Sci. Lett. 148, 13–26. |

[*] The asterisk sign means that the reference has been mentioned in our paper.

**Comment#2**: Secondly, in a journal like SE one expects a strong link between the presented results and the actual Earth, but instead the authors present a nondimensional parameter study with no attempt to make quantitative inferences or predictions regarding the actual Earth. This is in contrast to earlier studies on mantle convection with viscoelasticity, which the authors don't seem to know about since they don't cite them (Beuchert and Podladchikov, 2010; Harder, 1991; Moresi et al., 2003; Moresi et al., 2002) (also there are many papers on lithosphere & crust deformation that include viscoelasticity). The usual view is that in the mantle elasticity is unimportant at geological time scales because the viscoelastic relaxation time is short compared to the deformation time scale (viscosity/shear modulus ~1e21/1e11 ~ 300 years), while in the lithosphere, viscosity can be orders of magnitude higher so elasticity can be important. It is not clear that the authors reach this regime. As it is, this paper belongs more in a journal like Journal of Non-Newtonian Fluid Mechanics.

*Response: Using the dimensionless group (analogy) is useful for studying any fluid flow because the scale of different types of forces (by defining the Reynolds, Weissenberg and Rayleigh numbers) can be specified using this type of report and the results can be simply changed to the scale with real dimensions using the reference parameters. It is also so useful for experimental studies based on the analogy and making too smaller models (setups).*

**Comment#3**: Thirdly, the investigative method. If the authors want to demonstrate that nonlinear viscoelasticity is important then they need to show corresponding solutions for (i) nonlinear viscoelasticity (ii) linear viscoelasticity and (iii) viscous flow, and identify how and where they are different.

**Response**: *The mobility factor ($\alpha$) of Giesekus model helps us to adjust the nonlinearity degree of the model and it is discussed in the paper (refer to section 4.2.4 of revised paper). The model is linear for $\alpha = 0$ and the shear dependency is increased by increasing the mobility factor up to 0.5.*

**Comment#4**: Fourthly, they claim that they include viscous dissipation "for the first time" but in fact there are several 10s of papers on mantle convection that include this term – basically anything study that uses a compressible approximation. Here are just a few examples to get them started: (Balachandar et al., 1993; Glatzmaier, 1988; Jarvis and McKenzie, 1980; Leng and Zhong, 2008; Tackley, 1996).

**Response**: *Thank you for your comment. Your indication is correct. Actually, author's purpose is: the conjugated effect of nonlinear viscoelasticity and viscous dissipation is considered in the present study for the first time. This is corrected in the revised paper (refer to page 1 of revised paper).*

**Comment#5**: they claim that including variation of "g" through the mantle is an important new aspect of their study: actually the variation with depth is very small – only a few C2 SED Interactive comment Printer-friendly version Discussion paper % - as you can see in the widely-used PREM (Preliminary Reference Earth Model) (e.g. http://geophysics.ou.edu/solid_earth/prem.html). This is why it is almost always ignored.

**Response**: *In order to present a better simulation, we considered the variation of "g" in our CFD simulation. Actually, this effect is not considered in previous studies and it changes around 1.07% the maximum of velocity of vortices. This finding may be useful for future modeling to ignore or consider the changing in the gravitational acceleration with depth.*

*Best,*

*M. Norouzi*

---

## Author Comment (AC3) · 13 May 2016

**To the Reviewer #2**

Many thanks for the referee's valuable comments and his/her time spent in reviewing our paper (SE-2016-12). I would like to mention that in the revised paper, all the points have been taken into consideration. Following, I refer the comments made by the referee with their corresponding answers as italic font. The changes are highlighted in the revised paper (in yellow color).

1. Lack of literature review. The authors give a very rather incomplete review of previous work on mantle convection, for which an enormous amount of literature exists (interestingly they miss seminal work by McKenzie, Christensen, Tackley, Moresi, Solomatov and many others). I strongly suggest that they read up on the topic, for example by reading textbooks by Schubert, Davies, or some of the many review papers (by Tackley, Bercovici, Ricard, many of which are available online). As the current paper deals with viscoelastic convection, I had expected at least a complete review of existing work on this topic which is however also not the case, which shows a rather large ignorance towards previous work.

**Response:** *Due to the too number of literature on mantle convection, we decided to present only the best-known and related papers in the literature section of our paper. We have classified the collection of articles and books that we have access to them based on the main purposes of our work, i.e. Newtonian/non-Newtonian medium, the constitutive model for the viscosity, temperature- and depth dependency of viscosity model, geometry .... In addition, all the mentioned works by the reviewer have used the Maxwell constitutive model which is a linear viscoelastic model. It is important to remember that the Maxwell that is not able to predict the nonlinear viscosity. According to the reviewer's comment, we modified the paper by reporting these work in literature (refer to pages 4 & 5 of the revised paper).*

2. Gravity. Some of the other reviewers were a bit annoyed of your use of a depth-varying g, yet your polynomial does indeed reproduce the depth dependent effect of g on Earth, if I plot it on MATLAB, using the following lines:

» y=[1:3000]/1000; » g=-0.118*y.^6+ 0.602*y.^5 - 1.006*y.^4 + 0.6884*y.^3 - 0.3708*y.^2 + 0.167.*y - 9.846; » plot(g,y*1000), axis ij

The point is, however, that the variation of g within the Earth's mantle is very minor which your expression also shows, which is why it is usually assumed to be constant.

**Response:** *We checked data of Bullen (1939) and Dziewonski & Anderson (1981) and found that both of data are mostly similar with reasonable confidence. Both of data predict that the maximum variation of gravitational acceleration in mantle is about 6.8%. Although its variation is small, utilization of a depth-dependent gravity would lead to more realistic results. It is important to mention that Eq. (13) is dimensionless and we should not use it based on the meters of depth. In this Equation, $y = \tilde{y} / H$ is the dimensionless depth of the bottom plate and using the dimensionless depth (y), the formulation is completely correct.*

3. Employed constitutive relationships. All above mentioned papers employ linear viscoelasticity, for the simple reason that there is not much data to support the use of more complicated elasticity models for applications on the scale of a convecting mantle (apart maybe from using a Kelvin body for bulk deformation). One can of course come up with arbitrary complex constitutive relationships but if there is no data to back it up you are not modelling a problem that is geoscientifically relevant. It is unclear to me why the Giesekus model should be relevant for geoscientific applications and you don't give a justification for that which implies that your paper is simply irrelevant for geoscientific purposes.

**Response:** *This question is similar to comment#1 of the first reviewer that is answered in detail.*

4. Viscous dissipation Different than what you claim, viscous dissipation has been studied in the context of mantle convection models since at least the mid-80ies (please look at classical papers by Christensen and the above mentioned textbooks). Importantly, you should include adiabatic heating if you include viscous dissipation as they are of similar order of magnitude. You don't do that here.

**Response:** *Thank you for your valuable comment. In some studies, the effect of viscous dissipation on boundary condition has been considered as a small constant heat flux (instead of adiabatic boundary condition). Due to the small effect of viscous dissipation in scale of problem, the heat flux resulted from viscous dissipation is negligible so it can be removed*

*from the boundary condition. Therefore, in the viscous dissipation is just considered in the numerical domain (and not the boundary condition)*

5. Inertial terms Mantle convection is low Reynolds number fluid flow; yet you include inertial terms in your formulation. You can use that to model mantle convection but you would have to employ a very small timestep; moreover, you should first demonstrate that your code actually reproduces normal viscous convection for example by reproducing the Blankenbach benchmarks.

**Response:** *Thank you for your indication. The advection term is negligible in our CFD code and due to using the dimensionless form of governing equation; the time step is dimensionless so, the small dimensionless time cause no problem on convergence (The real time step is large enough). The code is also verified by comparing the results with previous solutions (refer to section 4.1 and Fig. 2) and the grid study is done to find the proper mesh and time step.*

6. Missing figures the uploaded pdf has no figure which makes it impossible to review it. Yet, from the text alone it is clear that this work is currently very far removed from being publishable in an Earth Science journal.

**Response:** *We are sorry for this problem. We sent the Figures separately during the first submission and unfortunately, there were not uploaded in the website by the office of Journal. After one week from submission, the Figures have been uploaded and in the revised version, we inserted the figures immediately after the manuscript to avoid any problem.*

*Best,*

*M. Norouzi*